

# A Hydrologically Based Model for Delineating Hazard Zones in the Valleys of Debris Flow Basins

Kaiheng Hu[1,2], Pu Li[1,2,3], Yong You[1,2], and Fenghuan Su[1,2]

[1]Key Laboratory of Mountain Hazards and Earth Surface Processes, Chinese Academy of Sciences, Chengdu 610041, China

[2]Institute of Mountain Hazards and Environment, Chinese Academy of Sciences & Ministry of Water Conservancy, Chengdu 610041, China

[3]University of Chinese Academy of Sciences, Beijing 100049, China

*Correspondence to*: Kaiheng Hu (khhu@imde.ac.cn)

**Abstract.** A hydrologically based model is developed for delineating hazard zones in valleys of debris flow basins. The basic assumption of this model is that the ratio of peak discharges of any two cross sections in a debris-flow basin is a power function of the ratio of their flow accumulation areas. Combining the advantages of the empirical and flow routing models of debris-flow hazard zoning, this hydrological model with minimal data requirements has the ability to produce hazard intensity values at different event magnitudes. The algorithms used in this model are designed in the framework of grid-based geographic processing and implemented completely on ArcGIS platform and a Python scripting environment. Qipan basin in the Wenchuan county of Sichuan province, southwest China where a large-scale debris-flow event occurred on July 11, 2013 was chosen as the test case for the model. The hazard zone identified by the model showed good agreement with the real inundation area of the event. The proposed method can help identify small hazard areas in upstream tributaries and the developed model is promising in terms of its application in debris-flow hazard zoning.

## 1 Introduction

Debris flow hazards are increasing owing to population growth and increasing constructions in the areas located in debris-flow prone mountainous regions all over the world, such as southwestern mountains in China (Cui et al., 2005; Ma et al., 2013), Alps Mountains in Europe (D'Agostino and Marchi, 2001), and western coastal mountains in America (Gartner et al., 2014). In addition to early warning and engineering structures for disaster prevention, debris-flow hazard zoning is an effective countermeasure and has attracted increasing attention in the recent decades (Dai et al., 2002; Fell et al., 2008). The main aims of hazard zoning are to identify potentially dangerous areas, classify different levels of the hazard, and produce digital hazard maps that are useful for disaster mitigation and also economical land use (Carrara et al., 1991; Hurlimann et al., 2008; Fell et al., 2008; Cui et al., 2011).

Many hazard assessment methods have been developed to determine the probability of occurrence and magnitude of hazards, or predict runout distance and inundation area of a future debris flow during the last few decades (O'Brien et al., 1993; Hungr et al., 1995; Iverson et al., 1998; Wei et al., 2003; Rickenmann, 2005; Quan Luna et al., 2013). These quantitative methods can be divided into empirical, flow routing, physically based analytical, and numerical methods (Hurlimann et al.,



2008). The latter two methods are the so-called dynamic models to estimate the propagation of debris flows through mass, momentum or energy conservation equations (Rickenmann 2005; Berti & Simoni, 2007). The analytical models regard debris flow as a mass point moving along a longitudinal slope, and calculate its runout distance and velocity at any position of the slope using Newton's second law (Rickenmann, 1990; Hungr, 1995; Rickenmann, 2005; Hungr and McDougall, 2009).

The numerical models treat debris flows as physical granular flow (Savage and Hutter, 1989), fluid flow (O'Brien et al., 1993; Wei et al., 2003), quasi-fluid flow (Denlinger and Iverson, 2001), and two-phase fluid-solid flow (Pitman et al., 2003; Pudasaini et al., 2005) on real topography, and solve their 1-D or 2-D depth-averaged governing equations through numerical approaches. Not only the analytical models, but also the numerical models need to be supplemented with additional constitutive laws, such as Coulomb grain frictional resistance law (Iverson, 1997; Hungr and McDougall 2009),

Bingham fluid visco-plastic rheological law (Laigle and Coussot, 1997; Malet et al. 2004), and Voellmy fluid frictional-turbulent rheological law (Rickenmann et al., 2006; Naef et al., 2006). Although the dynamic models can provide more accurate hazard intensity values, and produce elaborate hazard maps, the challenges of determining rheological parameters, the variability of controlling factors, and the sensitivity of rheological parameters and boundary conditions still remain.

In contrast, the empirical and flow routing methods require much less parameters than the dynamic methods. The empirical

methods are based on the empirical-statistical relationships among runout distance, cross-sectional area, inundation area, and magnitude of debris flows. Iverson et al. (1998) proposed a rapid, objective method of delineating lahar hazard zones in flat volcanic valleys based on two predictive empirical functions of inundated valley cross-sectional areas and planimetric areas with lahar volume. Berti and Simoni (2007) extended Iverson's model to non-volcanic debris flows in the Italian Alps using different empirical mobility relationships of the flow area and planimetric area with debris flow volume. The flow routing

methods delineate the debris-flow hazard zones by identifying process pathways or trajectories of debris flows from the source area to the deposition zone (Gamma 1999; Huggel et al., 2003; Wichmann and Becht, 2004; Horton et al. 2013). Grid-based "random walk" models coupled with a 2-parameter friction model and Monte Carlo simulation are used to generate the process pathways and predict runout distances (Wichmann and Becht, 2004; Horton et al. 2013). However, these empirical and flow routing methods do not take into account the probability of occurrence and magnitude of hazard,

and hence are applied to produce preliminary hazard maps (Hurlimann et al., 2008).

Inspired by the empirical model proposed by Berti and Simoni (2007) and the flow routing model by Huggel et al. (2003), we developed a hydrologically based model that can extract inundated hazard zones of different event magnitudes in valleys of debris-flow basins. Our model has the advantage of minimum data requirements, and it does not depend on the additional empirical-statistical relationships between the cross-sectional area and volume. Moreover, it can be easily implemented in

GIS software and output hazard intensity values at a local scale.

## 2 Methodology

### 2.1 Hydrological model

The model supposes a typical debris flow basin consisting of an outlet, a drainage divide (watershed boundary), and a

35 drainage network (fig.01). Once debris flows are initiated on hillslopes or in upstream tributaries, they will propagate downstream along the network, and accumulate to larger flow volume in the main channel. The flow discharge and average velocity are well-defined at the cross sections particular to each reach of the drainage channels (fig.01). The hazard intensity at the cross sections is described by a set of variables $(Q_i, U_i, H_i, A_{ci}, s_i)$. The peak discharge $Q_i$, average velocity $U_i$, and cross-sectional area $A_{ci}$ have the following relationship:



$$Q_i = A_{ci} \times U_i \tag{1}$$

The highest level of flow at the cross section depends on the cross-sectional topography and the peak discharge of a debris flow event (fig.01). If the peak discharge values at the cross sections of all the reaches are known, the flow depth and inundated cross-sectional area can be calculated in conjunction with the hydraulic relationships of open channel flows, for example, one-parameter Manning resistance law:

$$U_i = \frac{1}{n} H_i^{\frac{2}{3}} \times s_i^{\frac{1}{2}} \tag{2},$$

where $n$ is Manning resistance coefficient.

However, it is not practically feasible to measure the peak discharge of every cross section in a basin. Here, we propose a hydrological method to estimate the peak discharge of all the other cross sections from a known discharge value of a cross section on the basis of two simple hydrological assumptions: 1) trajectories of debris flows from the upstream tributaries to the downstream channel follow the drainage network; and 2) the ratio of the peak discharge at $i$th and $j$th cross sections has a relationship with the ratio of the flow volumes accumulated from their upstream contributing areas:

$$\frac{Q_i}{Q_j} = (\frac{V_i}{V_j})^m \tag{3}$$

where $V_i$ is the flow volumes across the $i$th cross section, and $m$ is the power exponent. The empirical value of $m$ ranges from 0.78 to 0.87 (Rickenmann, 1999).

The first assumption is straightforward because the flow path does not significantly deviate from the stream channel except in a very large-scale event. Although the second one may not hold true for a real debris flow event because deposition and erosion sometimes happen over a wide downstream channel and alluvial fan, the assumption is acceptable from a hydrological point of view. Moreover, it is reasonable to statistically expect a larger discharge at downstream sections than at upstream sections for debris flow events with a given return period. Furthermore, supposing that the ratio of $V_i/V_j$ is equal to $A_{fi}/A_{fj}$, where $A_{fi}$ denotes the flow accumulation area corresponding to the lowest pour point at the $i$th cross section (fig.01), the peak discharge $Q_j$ is calculated by:

$$Q_j = Q_i(\frac{A_{fj}}{A_{fi}})^m \tag{4}$$

The flow accumulation area is regarded as a kind of contributing area weighted by factors such as vegetation cover, rainfall, and landslide activity. Note that the flow accumulation area and the contributing area are equivalent only if rainfall and sediment supply are homogenous on the surface of the basin.

Two conclusions can be directly drawn from the second assumption: the peak discharge at the outlet is the largest; and once the peak discharge at one cross section is known, the discharge at any other cross section can be determined by the ratio of upstream flow accumulation areas. In reality, it is easy to obtain the value of flow peak discharge at a certain cross section by field survey or hydraulic methods. Next, we describe how to apply this model to map hazard zones in the valleys of debris-flow basins.





### 2.2 Algorithms

The tasks performed by the algorithms of this model consist of extracting primary drainage network, acquiring model parameters, computing flow depth and velocity, and delineating hazard zones. All the four steps are performed completely on a grid-based basin or a Digital Elevation Model (DEM).

1) To extract the drainage network: the most common single flow direction algorithm D8 is applied to extract a basin's primary drainage network from its DEM (fig.02). For each cell of the DEM grid, D8 algorithm directs the flow from that cell to one of its eight neighboring cells along the steepest slope (O'Callaghan and Mark, 1984). The flow direction grid is created from the DEM by assigning each of the cells a direction value (fig.02). The flow accumulation area of one cell is proportional to the number of cells that the cell receives flow from, and is obtained from the flow direction grid. The drainage network is drawn after setting a minimum value of flow accumulation area. The extracted network can be divided into many short reaches, each of which is represented by its starting or middle cells.

2) To acquire input parameters: the necessary parameters of the model include slope gradient, peak discharge, Manning resistance coefficient, and flow directions of all the reaches. The slope gradient of one reach is the mean value of the slopes of all the cells in the reach. If the peak discharge of one reach is known, the peak discharge of the other reaches can be estimated by Eq. (4). The resistance coefficient of the drainage network is determined by field-based work or hydraulic empirical relationship. The flow direction of one of the reaches is set to the D8 direction of its representative point or cell. The cross section of a reach is considered to be orthogonal to its flow direction. The flow width of a cell is equal to the cell size if the flow direction is vertical or horizontal, i.e., equal to 1, 4, 16, 64, and to 1.41421 times the cell size if the flow direction is diagonal, i.e., equal to 2, 8, 32, and 128 (fig.02).

3) To compute the flow depth and velocity: the inundated cross-sectional area $A_{ci}$ is calculated by the cross section topography and an initial value of the flow depth. The mean flow velocity $U_i$ is calculated by Eq. (2). If the product of $A_{ci}$ and $U_i$ is less than the peak discharge $Q_i$, the initial flow depth increases by a small value. Then, the process is repeated until the product of $A_{ci}$ and $U_i$ is larger than $Q_i$ (fig.03). Once the flow depth has been estimated, the widths on the right and left sides of the thalweg in the $i$th cross section are obtained by the number of the wetted cell on the right and left sides (fig.02). In general, the flow widths on the right and left sides of the stream line are unequal.

4) To delineate the hazard zone: If a person standing at a reach line faces downstream, the side on his right hand is defined as the right side of the reach, and the one on his left hand as the left side. The cells on the right and left sides of the reach, whose distances from the reach are smaller than the right and left widths, respectively, are regarded as falling into the inundation region (fig.02). The hazard zone is the collection of the inundation regions of all of the reaches.

### 3 Implementation

The hydrological model can be easily implemented in the ArcGIS platform (fig.04). Map Algebra and Hydrology toolsets can facilitate powerful hydrological analyses such as computing flow direction and accumulation area, delineating drainage network, and manipulating multiple grid operations. First, the sinks in the input DEM raster data are filled using the 'Fill' tool to ensure a continuous drainage network over the entire basin. Then, 'Flow direction' and 'Flow accumulation' tools are used to compute flow direction and accumulation area of each cell of the filled DEM. All cells for which the flow accumulation area is greater than a certain limitation comprise the drainage network; these are extracted by the Con function of the raster data in ArcGIS. Finally, the channel segments of the drainage network are numbered and vectorized by using the 'Stream link' and 'Stream to Feature' tools.





The 'Feature vertices to points' tool allows us to extract the middle or start points of a reach line. The cell containing these points is the representative cell of the reach. A resistance coefficient is assigned to each cell in the drainage network. The peak discharge of the cells in the network is calculated using Eq. (4) with the Map Algebra toolset if the peak discharge of one reach is given. The slope gradient is the mean value of the gradients of all the cells in a single reach, and obtained by the

Zonal Statistics function.

Tools for the extraction of the cross-sectional topography and computation of the flow depth and velocity are not available in the ArcGIS platform. Fortunately, ArcGIS provides additional geoprocessing functionality and extension through the ArcPy site package; this allows an advanced developer to perform more complex geographic data analysis in Python scripting environment. The powerful mathematic packages in Python such as NumPy and SciPy can be easily integrated into our

hydrologic computation system with ArcGIS geoprocessing functionality. First, the raster datasets of the peak discharge, slope gradient, resistance coefficient, and flow direction are converted into NumPy array objects in Python. Then, the flow depth, flow velocity, and cross-sectional area are solved iteratively according to the flow chart shown in Fig. 3. When the flow depth and velocity solutions for each channel reach are achieved, the right and left widths along the reach line are determined and exported into ArcGIS as buffer parameters.

The "buffer" tool of ArcGIS is used to delineate the inundation zone or hazard zone along the main channel. As mentioned above, the regions within the right width on the right side of each reach and the left width on its left side are merged into a single area that is regarded to be the inundation zone of debris flow in the valley. Finally, the merged inundation polygon is smoothed by the "Smooth" tool.

**4 Model testing**

The case study is of Qipan basin that is located in Wenchuan County, Sichuan Province of Southwestern China, and a tributary of Min River (fig.05). The catchment area is 52.4 km$^2$, main stream is 15.8 km, and average channel gradient is 170‰. The elevation of the basin ranges from 1320 m to 4360 m a.s.l. The basin is in VIII intensity seismic region. The main channel is nearly orthogonal to a large NE-trending fault at the mouth of the basin. The cropped rocks in Qipan basin

mainly consist of the Sinian dolomite, and Proterozoic diorite and granite. The sediment source of the debris flows includes weathered material and soil mantled on bedrocks at hillslopes.

Small-scale debris flow events, with peak discharge less than 100 m$^3$/s, occurred in the basin before the Wenchuan earthquake on 12 May 2008. These debris flows were not hazardous, and hence more than 1000 people reside on the both sides of the main channel in the downstream valley. The Wenchuan earthquake triggered many landslides and avalanches

over the basin and two small dammed lakes in the main stream. Massive loose materials from these seismic hazards significantly increased the frequency and magnitude of the debris flows. A large-scale debris flow event was triggered by a heavy rainstorm that occurred between 8–12 July 2013. The cumulative rainfall was 118 mm, which is near the record high. The event occurred at 3:00 a.m. on 11 July when the rainfall amount reached at a level of 111.6 mm. The event duration was approximately 30 min. A total of 15 people died or were missing, 350 buildings were completely destroyed, and more than

2000 buildings were buried in this event (Zeng et al, 2014). According to a field survey after the event, the peak discharge was 1745 m$^3$/s, and the average flow depth was 5 m at the cross section K1, where the slope gradient was 0.11 and the Manning coefficient was 1.0/12.0 (fig.05). The average velocity was estimated to be 11.64 m/s from Eq. (2).

The event on 11 July was chosen as the test case for our model. The DEM of the basin was discretized from 1:10000 contour maps with a resolution of 5 m × 5 m. The peak discharge was set to 1745 m$^3$/s at the reach that intersects with the K1 cross



section. Then, the peak discharge values at all the reaches were calculated by multiplying 1745 m$^3$/s with the ratio of the flow accumulation area. Applying the algorithms and implementation methods in Sections 2 and 3 to the case, the hazard zone could be extracted as shown in Figure 6. It was noted that the boundaries of the hazard zone are discontinuous at the binding point of the two neighboring reaches as mentioned above. The Smooth tool was used to smooth the piecewise

boundaries. The DEM resolution and topographic change was found to have significant influence on the computation result. It was obvious that the inundation width of the downstream reaches is bigger than that of the upstream ones. The hazard zone is wider at the junction of two tributaries. An advantage of our model is that small areas can also be identified as the hazard zone at narrow upstream channels.

In order to test the validity of our model, the obtained hazard zone result was compared with the real inundation area of the

event (fig.07). The hazard zone had an acceptable agreement with the inundation area. Most damaged or destroyed buildings were within the hazard zone.  However, the hazard zone slightly underestimated the inundation area at some downstream reaches, which was likely due to the blockage of buildings or the interpolation accuracy of DEM. Moreover, the hazard zone at small branches was not inundated by this debris flow. It is reasonable that the obtained hazard zone is just a potential region that could be subjected to debris-flow hazard.

### 5 Discussion

In the Qipan example, some simplifications were made such as for the assignment of the resistance coefficient and computation of the accumulation flow area. In reality, the coefficient is not identical for different reaches. Assigning a uniform resistance coefficient to all the reaches has an influence on the resulted hazard zone. In order to obtain finer results,

non-uniform resistance coefficients are required to be assigned to different reaches, which can be acquired by a detailed channel survey. When calculating the flow accumulation area, the sediment supply and rainfall are considered to be uniform; hence, the accumulation area is equal to the contributing area. A more precise model should consider the influence of the spatial distribution of the sediment supply and rainfall on the flow accumulation area. In the future, the computation of the flow accumulation area will take into account the non-uniformity of the rainfall and the sediment supply. Essentially, the

simplifications do not affect the hydrological model though they affect the accuracy of the obtained hazard map.

In addition, a special method is used to handle unconfined flow when the valley at some downstream reaches is so wide that the overflow occurs. Equation (2) is suitable for confined channel flow but cannot handle unconfined flow. Berti and Simoni (2007) applied an empirical relationship between the debris flow volume and mean depositional thickness for treating unconfined flow. Unlike them, we adapted an empirical equation between the mean flow velocity and the discharge

presented by Rickenmann (1999). In the case of unconfined flow, i.e. when Eq. (2) has no solution, the empirical equation is used in the algorithm of computing the flow depth and velocity.

### 6 Conclusions

In most cases, limited data such as peak discharge, flow depth, or mean velocity of debris flows at a cross section can be

easily obtained. In this paper, the limited data are utilized to delineate the hazard zone in the valleys of a debris flow basin based on a reasonable hydrological relationship between the cross-sectional peak discharge and flow accumulation area. We proposed that the ratio of the peak discharges of any two cross sections in the basin satisfies a relationship with the ratio of their flow accumulation areas as shown in Eq. (4), and also developed an objective hydrologically based model of debris-flow hazard zoning. The hydrological model requires only the values of the peak discharge and the resistance coefficient at



one reach of a debris-flow basin, and is capable of producing hazard intensity values corresponding to different event magnitudes. The model and its algorithms such as extracting the primary drainage network, acquiring model parameters, computing the flow depth and velocity, and delineating hazard zone are completely implemented in the ArcGIS platform and Python environment. We applied the model to Qipan basin where a devastating event occurred on 11 July 2013. The overlay

5   of the aerial photo with the obtained hazard zone shows that most of damaged and destroyed buildings in this event were in the hazard zone. Comparison of the obtained hazard zone with the real inundation area of the event demonstrates that the model has the ability to capture a hazardous area at an acceptable level. Future improvements such as considering non-uniform distributions of resistance coefficient, rainfall, and sediment supply will make the model more feasible.

**Acknowledgements**

This work has been supported by the Key Research Program of the Chinese Academy of Sciences (Grant No. KZZD-EW-05-01), the National Basic Research Program of China (973 Program) (Grant No. 2015CB452704), and the Open Foundation of Key Laboratory of Mountain Hazards and Earth Surface Processes, Chinese Academy of Sciences.



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





**Figure Captions**

Figure 1. Schematics of the hydrological model (the drainage network is composed of many small reaches (blue lines), each of which can be represented by a lowest pour point. The peak discharge $Q_i$, inundated cross-sectional area $A_{ci}$, cross-sectional average velocity $U_i$, channel bed slope $s_i$, peak flow depth $H_i$ are well-defined at the $i$th cross section through the $i$th pour point. The flow accumulation area $A_{fi}$ corresponding to the $i$th point is supposed to be equal to its contributing area, i.e., the subbasin area (green polygon)).

Figure 2. Illustration of the model's algorithms (the red arrows denote the flow direction, and (a) the definitions of eight directions of D8 algorithm; (b) the right and left widths of a cross section; the grey area denotes the inundated cross-sectional area and depends on the maximum flow depth.)

Figure 3. Flow chart of the iteration algorithm for computing the flow depth and velocity

Figure 4. Framework of the model's implementation in ArcGIS and Python

Figure 5. Location map of Qipan basin (K1 section is the investigated cross section of the event as on 11 July 2013)

Figure 6. Hazard zone of the 11 July event identified by the methodology (maximum length of each reach is set to 50 m and the distance between the two representative points is less than 50 m.)

Figure 7. Comparison of the obtained hazard zone and real inundation area of the event (overlaid base map is the aerial photo taken shortly after the debris-flow event occurred, which is provided by Geomatics Center of Sichuan Province. Red line represents the outline of the hazard zone.)




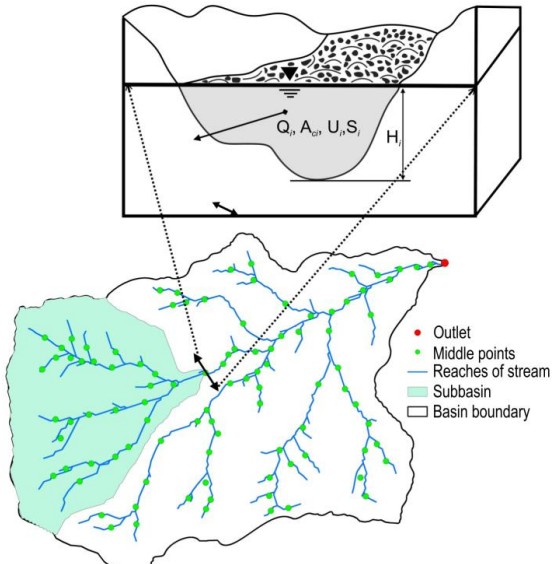

Figure 1



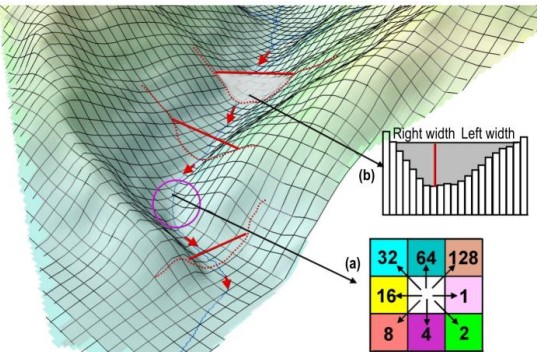

Figure 2





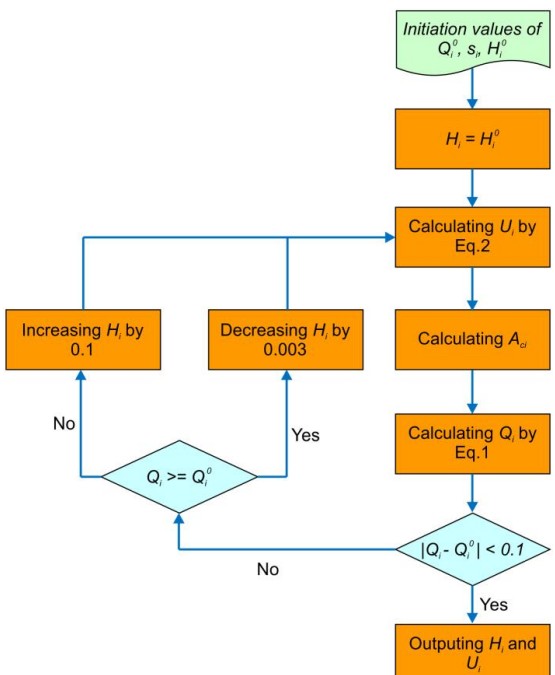

Figure 3




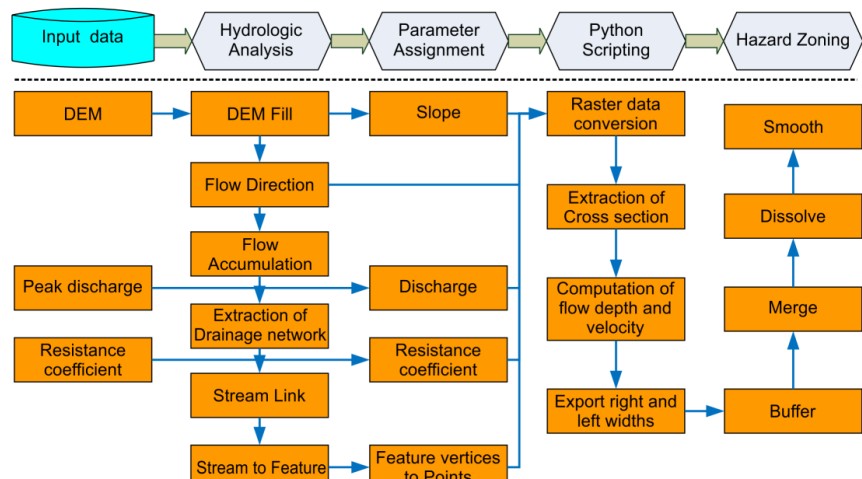

Figure 4



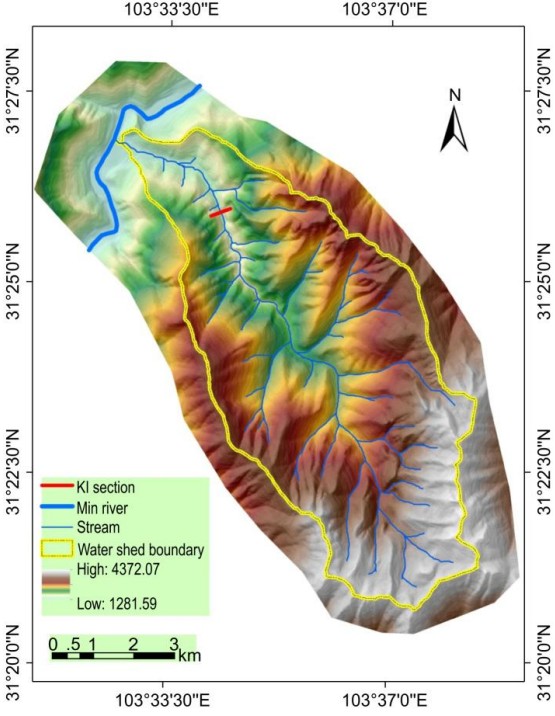

Figure 5





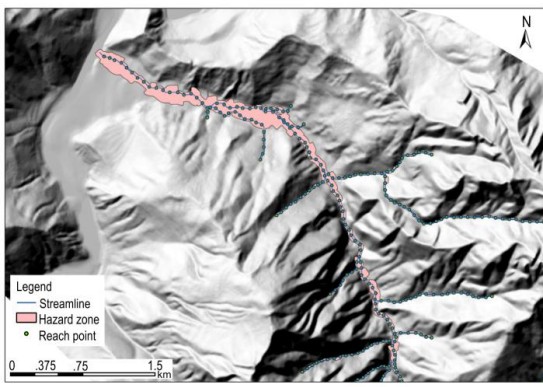

Figure 6



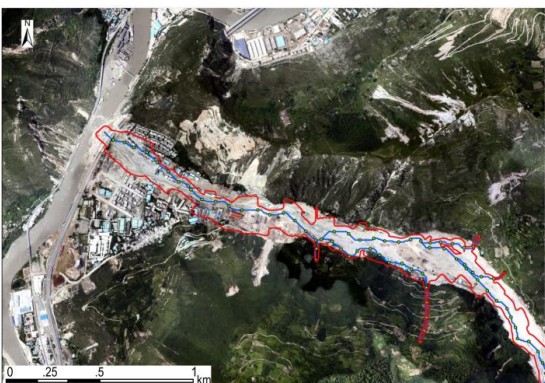

Figure 7

