# Peer review of "A Hydrologically Based Model for Delineating Hazard Zones in the Valleys of Debris Flow Basins"

_Natural Hazards and Earth System Sciences, 2016_

## Referee Comment (RC1) · Anonymous Referee #1 · 8 Mar 2016

General comments:

The manuscript (ms) aims to come up with as much hydrological information as possible in data scarce or even data absent situations of debris flows. The overall aim is to delineate debris flow hazards in basins. This aim should be applauded. Debris flows are notorious underdetermined as it is very hard to get measurements on regular basis of these events. Every approach to try to describe hydrology in such cases deserves positive attitude. I do not fully agree with the description of the aim of the work nor the terminology the authors used. To me, the ms proposes a geomorphological/topographical approach to mimic hydrology based basically on DEM information. The aim is to delineate debris flow hazards based on this approach. The proposed method is very simple and straightforward: use areal ratios to estimate discharge through a cross section. Only one discharge measurement is needed. Basic

assumptions are uniform flow generation, flow hydraulics and sediment supply. Simplicity in this case can – I think – be an advantage.

However, the authors have no data to test their method, nor used published data sets to test/validate their approach. Additionally, the paper lacks a scientific approach. The authors present the results of their approach but show/present no effort to really analyse/evaluate/proof/falsify their approach. So, although the approach could make sense, the authors 'only' present the single run of their model and then not even with hydrological information (the innovation) but the hazard map (final aim). In other words, nothing is proven and that leaves the reader disappointed. Therefore, I have to reject this ms for publication. In my opinion the paper could have some potential but in current form is far from complete/finished/publishable, it is a start at the most.

In the following I give a) a list of possible improvements and b) a set of minor items (not complete) to think over or address. I hope the authors will follow up, significantly improve their scientific method, analysis and discussion of their approach and resubmit such a ms. - Reformulate the aim/method. This is no hydrology-based model but a topography based approach to make up for the lack of hydrology information. - Define a method to calibrate/validate/proof your model (see next point) - E.g. as you have no data, all you have 1 one description, so the field testing is not really a scientific proof. I suggest you try to find data, preferably already published, that can serve as kind of benchmark verification of your approach. So you run your model on that. - Another way forward could be, to define a simple hypothetical (synthetic) catchment, run full numerical model and then compare outcomes with your approach. Evaluate strong and weak points. - Also elaborate on this one field example you have. Show/explain your method of deriving your discharge estimate etc (see also under "other points".). - Perform at least some kind of sensitivity for your approach, e.g. the influence of the DEM resolution (which you write yourself has influence (of course, it influences everything you need in your model, such as the cross sections you derive), but also quantify the effect of the Manning coefficient. Just writing it has influence is not enough.

Other points: - P1, L23: hazard should be Risk - P1L27: countermeasure -> measure - P1L27-30: you only define one aim (hazard zoning). Digital maps are no aim, classify different levels of hazard is not what you do. - P1L34: I do not get yoyr model-type classification system. - P2L37: well-defined? In your DEM based approach there is no such thing as well-defined - P3Eq.1: define all subscripts - P3L9-27: I understand you describe the assumptions very well, but please also test these assumptions in the following research. That is really missing. - P3L22: Do you have proof that volume ratios can be approximated with areal ratio? Can you do some back-of-envelope cal-culations. And why not allowing exponent m to change? In other words, $(V/V)^m = (A/A)^n$ (and then estimate what n should be). This could be done perhaps with a synthetic analysis using GIUH approach? - P4 section 2.2. Don't you agree "algoritms" is a too big word for 1 formula you are using? - P4L26-29: unnecessary explanation. Left and right bank of stream are always defined like that in geomorphology - P5L6-17: It could be interesting if you give your scripts in an appendix so to evaluate this as well and others to also test/use it. - P5 section 4: this is not model testing. You 'only' apply it without critical evaluation. - P5L35: "According to field survey". Explain how performed, which method you followed. And explain how you can get 1745 m3/s with 12 m/s flow velocity. - P5L36-37: Discuss overtopping (this discharge is more than bank-full discharge, which you use in your approach). Also discuss effect of sediment scouring and additional bank erosion with such flow velocities - P6L5: of course DEM resolution is important. Please elaborate on this, show sensitivity and discuss effects. - P6L10: Define "acceptable agreement". - P6 Section 5 Discussion: This is not a discussion on your results. You only point to some problems without really relating it to your model results.

---

## Author Comment (AC1) · 11 Apr 2016

We are thankful for the referee's valuable comments that are very helpful to improve our MS substantially. We are also delighted to see that the basic ideas of our methodology are appreciated by the referee. Our aim, as pointed out by the referee, is to extract debris-flow hazard zone in the case of no sufficient hydrological information. We used the term of "hydrologically based" to reflect the content of basin hydrological analysis and simple hydrological relationship. But, with the comments, we realize that the term may be easily misunderstood as regular hydrological model such as infiltration, rainfall, water balance etc. We think "geomorphological/topographical" cannot fully depict our proposed method even though key elements of our method are related to DEM. We try to find a better term for our approach.

[Figure]

We do not want to go through all the comments. We will take them very seriously in our next research. The most important is we agree with the referee that our model needs more scientific proof and cases to support. We are going to apply published cases, numerical models and even real hydrological models to validate/calibrate /evaluate our approach. We hope our works can be accepted by scientific community of natural hazards after great effort.

---

## Referee Comment (RC2) · M. Mergili (Referee) · 17 May 2016

The authors present a quite straightforward approach for debris flow hazard zoning, requiring a minimum of input information. The approach presented is interesting, and the manuscript is generally well structured and illustrated. However, part of the content remains at a very basic level, so that I cannot recommend the manuscript in its present form for publication in a highly-ranked journal such as NHESS. Therefore I recommend some major revisions according to the comments and suggestions given below.

The authors are welcome to contact me at martin.mergili@univie.ac.at in case they disagree with my comments or if they wish to further discuss the one or the other issue.

1. Some standard GIS functions are described in too much detail (e.g. Page 4, Lines 5-

19 and 34-39). This is not necessary and distracts the reader from the more innovative parts of the manuscript. Fig. 4 does the work of showing which ArcGIS raster functions are used, none of these functions has to be explained in detail.

2. Is it valid to assume $V_i/V_j = A_{fi}/A_{fj}$? Are there some references proving this?

3. What was the value of m chosen for the test analysis?

4. The model testing in general represents a huge potential for improvement. More thorough testing is absolutely necessary to make the manuscript acceptable for publication. (i) The model should be run with different parameter settings (e.g., m) in order to explore the uncertainties and the parameter sensitivity. (ii) A more quantitative evaluation of the results is necessary ("... acceptable agreement ..." is not sufficient). (iii) The test case consists in the reproduction of an observed event. It could be interesting to – after optimizing the parameters with the observed inundation areas - demonstrate some "real" hazard zoning i.e. by considering peak discharges of events with different recurrence intervals and deriving an annual probability of impact for each pixel. The results should be evaluated in another test area. Such an effort could be really interesting to the audience of NHESS and add a lot of value to the article. If there are no frequency-magnitude relationships known for the area, a scenario-based analysis for different debris flow magnitudes could be demonstrated.